# A Novel Chaotic Image Encryption Scheme Armed with Global Dynamic Selection

**DOI:** 10.3390/e25030476

**Published:** 2023-03-09

**Authors:** Xin Chen, Qianxue Wang, Linfeng Fan, Simin Yu

**Affiliations:** College of Automation, Guangdong University of Technology, Guangzhou 510006, China

**Keywords:** chaotic encryption, equivalent key, ciphertext feedback, dynamic selection

## Abstract

Due to the equivalent keys revealed by a chosen-plaintext attack or a chosen-ciphertext attack, most of the existing chaotic image encryption schemes are demonstrated to be insecure. In order to improve security performance, some scholars have recently proposed the plaintext-related chaotic image encryption scheme. Although the equivalent effect of a one-time pad is achieved, an additional secure channel is required to transmit the hash values or other parameters related to the plaintext before the ciphertext can be decrypted at the receiving end. Its main drawback is that an absolutely secure channel is needed to transmit the information related to the plaintext, which is not feasible in practical applications. To further solve this problem, this paper proposes a chaotic image encryption scheme based on global dynamic selection of a multi-parallel structure. First, a chaotic sequence is employed to dynamically select DNA encoding rules. Secondly, the permutation with a multi-parallel structure is performed on the DNA-encoded matrix, and the DNA decoding rules are dynamically selected according to another chaotic sequence. Finally, the diffusion rules obtained by the ciphertext feedback mechanism are introduced to determine the dynamic diffusion. Compared with the existing local dynamic encryption schemes, the main advantage of this scheme is that it can realize global dynamic selection, so as to ensure that there is no equivalent key, and it can resist the chosen-ciphertext attack or chosen-plaintext attack and does not need an additional secure channel to transmit parameters related to plaintext, which is practical. A theoretical analysis and numerical experiments demonstrate the feasibility of the method.

## 1. Introduction

With the development of today’s science and technology, all aspects of people’s lives have undergone informatization [1,2]. As a medium in the information age, images can directly convey the message that people want to express [3]. Information technology has an increasing impact on personal privacy, medicine, and social interaction. Once the important information in the image is intercepted or tampered with by the attackers, the damage caused cannot be ignored [4,5]. Therefore, it is very important to protect the safe transmission and reception of image data. In order to ensure the security of digital images, researchers have proposed many image encryption methods based on different technologies [6,7,8].

Generally speaking, image encryption algorithms are mainly divided into two operations: permutation and diffusion [9]. Permutation changes the position of pixels, and its main purpose is to break the correlation between the adjacent pixels of an image. Diffusion changes the pixel value of each pixel in a specific way to achieve the purpose of protecting image information. In essence, diffusion is the operation of changing the pixel value [10,11]. Chaotic systems have the characteristics of pseudo-randomness, initial value sensitivity, parameter sensitivity, and unpredictability [12], which can be applied in the field of image encryption [13,14]. The chaotic sequence generated by the chaotic system iteratively participates in the permutation and diffusion, so as to improve the security of the cryptographic system. Due to the characteristics of chaotic systems and the high adaptability of image encryption technology, chaotic image encryption technology has gradually attracted the attention of researchers [15,16].

With the deepening of research on chaotic image encryption technology, chaotic image encryption algorithms are mainly divided into three categories [17,18]. The first type is encryption by a self-synchronous stream cipher, which does not require an additional secure channel and is practical [19,20,21,22,23]. The second type is encryption related to plaintext; this scheme has no equivalent key, and it is difficult to crack, but the premise is that additional parameters, such as the hash value, need to be assumed to be transmitted through an additional secure channel, which is not practical. Although Chai et al. improved on this basis by embedding key parameters into the cipher image and transmitting it together with the latter cipher image, it can not resist cropping attacks well [24]. The third type is a hybrid encryption that combines other technologies, such as local dynamic encryption, which does not consider the equivalent key and other situations and has difficulty resisting chosen-plaintext or chosen-ciphertext attacks. For example, Xian proposed a fractal sorting matrix and its application in chaotic image encryption [25]. The pixel positions in each round of permutation are the same, and the diffusion is orderly, which reduces the dynamics and randomness of the scheme. These make the scheme less secure [26,27,28].

In order to make the chaotic image encryption scheme dynamic and flexible, some encryption links with parallel structures are considered in permutation and diffusion [29,30,31,32]. In 2018, Yin proposed a chaotic image encryption scheme based on a breadth-first search and dynamic diffusion [33]. In 2019, Li proposed a chaotic image encryption method with orbit perturbation and dynamic state variable selection mechanisms [34]. In the same year, Meysam proposed a chaotic image encryption scheme based on a polynomial combination of chaotic maps and dynamic function generation [35]. In 2021, Wu proposed a plaintext-related dynamic key chaotic image encryption method [36]. These schemes use local dynamic selection to improve flexibility and multi-selectivity in the encryption [37,38,39].

In order to further improve the flexibility of a chaotic image encryption scheme, this paper proposes a chaotic image encryption scheme based on global dynamic selection to realize the dynamic selection of bit-level, pixel-level, and image-level encryption by designing a multi-parallel structure. First, DNA encoding rules are dynamically selected according to the chaotic sequence. Then, the DNA-encoded matrix is dynamically permuted. Next, the DNA decoding rules are dynamically selected according to the chaotic sequence. Finally, dynamic diffusion is performed by the diffusion rules obtained by different locations. The main feature of this scheme is that it can realize global dynamic selection, so as to ensure that this scheme cannot crack the equivalent key and can resist chosen-plaintext attacks and chosen-ciphertext attacks. Our scheme does not need to use an additional secure channel to transmit parameters related to plaintext, so it is practical. The simulation results and the performance analysis show that the designed scheme has high security and good performance indicators.

The remainder of this research work is organized as follows. The overall framework of the scheme and the basic theory of 2D-LSM, DNA coding, dynamic permutation, and dynamic diffusion are given in Section 2, while the security of the scheme is analyzed theoretically in Section 3. Simulation experiments and performance analysis are detailed in the Section 4. This article ends with a Conclusion section in which the contributions are summarized (Section 5).

## 2. Chaotic Image Encryption Scheme

This paper proposes a chaotic image encryption scheme based on global dynamic selection. Its design idea is to build a multi-parallel structure, and its main feature is to realize dynamic selection through the multi-parallel structure design in encryption. First of all, all encryption processes in this scheme are called “global”. Secondly, for an image to be encrypted, the orders of magnitude of each encryption process are the bit level, pixel level, and image level, respectively, i.e., processing an image at the bit level, pixel level, and image level can also be called global. Dynamic means that when encrypting the same order of magnitude, the encryption rules executed on the same process will change instead of being fixed. Specifically, at the bit level, it is realized through the DNA encoding process and the DNA decoding process. For every two adjacent bits, their rules of DNA encoding and DNA decoding are different. The pixel level is completed by dynamic diffusion, and the diffusion equation performed by every two adjacent pixels is different. The image level is realized by dynamic permutation; for the same image, the first round and the second round of the permutation are determined by the calculated permutation rule value. The permutation rule value is not fixed, and each number corresponds to a rule. The so-called parallel structure means that within the same encryption process, there are multiple available encryption rules. For example, in dynamic diffusion, each pixel will have two diffusion methods, but the specific implementation of the diffusion method can only be known after the diffusion rules are determined. However, encryption with a non-parallel structure often has only one rule to perform encryption, and the encryption method has been fixed. By designing a parallel structure, each encryption process has multiple parallel encryption rules, and the specific process rules in encryption are selected by the chaotic sequence, permutation rule value, and diffusion rule value.

Different from the existing schemes, the main feature of this scheme is that, even if it is not related to the plaintext, the equivalent key cannot be cracked within a limited number of years. The existing dynamic encryption is mainly local dynamic encryption. Local dynamic encryption realizes dynamic selection in some processes of encryption. We propose an encryption scheme with global dynamic selection to achieve dynamic selection in all processes of encryption. At the same time, it also realizes dynamic selection from the three aspects of the bit level, pixel level and image level for the first time. In the process of DNA encoding and DNA decoding, DNA encoding and decoding rules are selected according to the chaotic sequence to realize dynamic selection at the bit level. In the dynamic permutation, according to the permutation rule value, the permutation method is dynamically selected to realize the dynamic permutation at the image level. In dynamic diffusion, the diffusion equation of each pixel is selected through the diffusion rule value to achieve dynamic selection at the pixel level.

As the number of encryption rounds increases, the permutation method and diffusion equation performed by the first round and the second round of encryption will change due to the permutation rule value and the diffusion rule value. The encryption rules executed in different rounds are different. This reflects the characteristics of dynamic selection. The global dynamic selection feature of the scheme is reflected in two aspects:1.All elements of an image can be classified into bit level, pixel level, and image level. This scheme dynamically selects a specific encryption from these three levels to encrypt the image.2.Using the chaotic sequence and the designed multi-parallel structure, the design concept of dynamic selection is reflected in the encryption rules that need to be selected and executed for each process.

### 2.1. Scheme Description

The block diagram of the proposed image encryption scheme is shown in Figure 1. Without loss of generality, an encrypted object can be reduced to an image *P* of size L=M×N, represented by a two-dimensional (2D) eight-bit integer matrix $P={\left\{p(i,j)\right\}}_{i=1,j=1}^{M,N}$; the final cipher image obtained after encryption through this scheme is $C={\left\{c(i,j)\right\}}_{i=1,j=1}^{M,N}$. Each piece of two-dimensional image data can also be written as a one-dimensional (1D) array scanned in raster order (left to right, top to bottom). For example, $P={\left\{p(i)\right\}}_{i=1}^{L}$. In Figure 1, the single-throw switch $K_{1}$ is turned on first, and, after entering the plain image *P*, $K_{1}$ is disconnected, and the double-throw switch $K_{2}$ is connected to position one. The image, after the first round of encryption, is fed back to the input for the second round of encryption. Then, connecting $K_{2}$ to position two outputs the cipher image. In Figure 1, *I*, *S*, *E*, *D*, $R_{S}$, and $R_{D}$ are the DNA-encoded matrix, permutation matrix, DNA-decoded matrix, diffusion matrix, permutation rule value, and diffusion rule value of the encrypted image.

In this scheme, the sub-block diagram of “2D-LSM” is a two-dimensional chaotic system proposed by Hua et al. [17], and the mathematical expression of the iteration function is: (1)$$\left\{\begin{array}{c} h\left(i+1\right)=\cos\left(4\alpha h\left(i\right)\left(1-h(i)\right)+\beta \sin\left(\pi w(i)\right)+1\right),\\ w\left(i+1\right)=\cos\left(4\alpha w\left(i\right)\left(1-w(i)\right)+\beta \sin\left(\pi h(i)\right)+1\right), \end{array}\right.$$
where $h\left(i\right),w\left(i\right)\in \left[0,1\right]$. The system is in a chaotic state when $\alpha ,\beta \in \left[1,100\right]$. This chaotic system has a total of four key parameters $\left\{h\left(0\right),w\left(0\right),\alpha ,\beta \right\}$, The system is iterated by the first set of initial key parameters $\left\{h_{1}\left(0\right),w_{1}\left(0\right),\alpha_{1},\beta_{1}\right\}$ to obtain the chaotic sequences *A* and *B*. The system is iterated by the second set of initial key parameters $\left\{h_{2}\left(0\right),w_{2}\left(0\right),\alpha_{2},\beta_{2}\right\}$ to obtain *X* and *Y*.

This scheme realizes bit-level dynamic selection through sub-block diagrams of “Dynamic DNA encoding *I*” and “Dynamic DNA decoding *E*”. The process of DNA encoding is to divide each eight-bit binary pixel of image *P* into four two-bit binary bit pairs, according to the corresponding value in *A*. The DNA encoding rule to be executed is dynamically selected to realize the DNA encoding from the number matrix to the symbol matrix. The DNA-encoded matrix *I* is obtained. The process of DNA decoding is the opposite of that of DNA encoding. According to the corresponding value in *B*, the decoding rule is dynamically selected to decode each of the four symbols into an eight-bit binary pixel, and the DNA decoding matrix *E* is obtained.

The sub-block diagram of “Dynamic permutation *S*” realizes the dynamic selection of different permutation methods for the DNA-encoded matrix *I* and introduces the permutation rule value $R_{S}$ to select and execute four different permutation methods. $R_{S}$ is determined by the symbol values of the four corners of the matrix *I*, and the matrix after dynamic permutation is denoted as *S*.

Through the sub-block diagram “Dynamic diffusion *D*”, the dynamic diffusion of pixels in different positions of the DNA-decoded matrix *E* is realized. According to the diffusion rule value $R_{D}$, select the specific execution rule from two different diffusion rules, and the image after diffusion is *D*. $R_{D}$ is obtained by feedback from the DNA-decoded matrix *E*, chaotic matrix *X* and diffusion matrix *D*. *D* is obtained by substituting the feedback of the DNA-decoded matrix *E*, the chaotic matrix *Y*, and the diffusion matrix *D* into the diffusion equation determined by $R_{D}$.

$K_{1}$ is a single-throw switch, which is used to cut off or connect the plaintext input encryption system; $K_{2}$ is a double-throw switch, which is used to connect the feedback loop when $K_{2}$ is at one and to connect the ciphertext output branch when $K_{2}$ is at two. The ciphertext output branch is used to output the final encrypted cipher image.

### 2.2. The Encryption Process

This section will introduce the encryption process of this scheme, and the detailed process is as follows:

(1) Initialization: Iterate the 2D-LSM chaotic system 800 times from the initial conditions $h_{1}\left(0\right),w_{1}\left(0\right)$ with the control parameters $\left\{\alpha_{1},\beta_{1}\right\}$ to avoid the transient effect in the initial iteration, and then iterate it 4L more times to obtain the two state sequences ${\left\{h_{1}\left(i\right)\right\}}_{i=1}^{4L}$ and ${\left\{w_{1}\left(i\right)\right\}}_{i=1}^{4L}$. Quantize them to two eight-bit integer sequences $A={\left\{a\left(i\right)\right\}}_{i=1}^{4L}$ and $B={\left\{b\left(i\right)\right\}}_{i=1}^{4L}$ via
(2)$$\left\{\begin{array}{c} a\left(i\right)=\;\mathrm{mod}\;\left(fix\left(h_{1}\left(i\right)\times 10^{10}\right),8\right)+1,\\ b\left(i\right)=\;\mathrm{mod}\;\left(fix\left(w_{1}\left(i\right)\times 10^{10}\right),8\right)+1, \end{array}\right.$$
where fix(·) is the rounding down function, mod(·) is the modulo operation, and i=1,2,⋯,4L.

Iterate the 2D-LSM chaotic system 800 times from the initial conditions $h_{2}\left(0\right),w_{2}\left(0\right)$ with the control parameters $\left\{\alpha_{2},\beta_{2}\right\}$ to avoid the transient effect in the initial iteration, and then iterate it *L* more times to obtain two state sequences ${\left\{h_{2}\left(i\right)\right\}}_{i=1}^{L}$ and ${\left\{w_{2}\left(i\right)\right\}}_{i=1}^{L}$. The one-dimensional arrays ${\left\{h_{2}\left(i\right)\right\}}_{i=1}^{L}$ and ${\left\{w_{2}\left(i\right)\right\}}_{i=1}^{L}$ can also be written as two-dimensional eight-bit integer matrices, ${\left\{h_{2}\left(i,j\right)\right\}}_{i=1,j=1}^{M,N}$ and ${\left\{w_{2}\left(i,j\right)\right\}}_{i=1,j=1}^{M,N}$, by scanning them in raster order. Quantize them to two eight-bit integer sequences $X={\left\{x\left(i,j\right)\right\}}_{i=1,j=1}^{M,N}$ and $Y={\left\{y\left(i,j\right)\right\}}_{i=1,j=1}^{M,N}$ via
(3)$$\left\{\begin{array}{c} x\left(i,j\right)=mod\left(fix\left(h_{2}\left(i,j\right)\times 10^{10}\right),255\right)+1,\\ y\left(i,j\right)=mod\left(fix\left(w_{2}\left(i,j\right)\times 10^{10}\right),255\right)+1, \end{array}\right.$$
where fix(·) is the rounding down function, mod(·) is the modulo operation, and i=1,2,…,M;j=1,2,…,N.

(2) DNA encoding: The switch $K_{1}$ is closed, so that each pixel $p\left(i\right)\left(i=1,2,\ldots ,L\right)$ in *P* corresponds to four two-bit binary pairs. Then, use the $a\left(4i-3\right),a\left(4i-2\right),a\left(4i-1\right),a\left(4i\right)$ column encoding rules in Table 1 to transform them into DNA symbols. Because one pixel corresponds to four symbols, the matrix *P* will be reshaped into a symbol matrix $I={\left\{i\left(j,k\right)\right\}}_{j=1,k=1}^{M,4N}$ with *M* rows and 4N columns, consisting only of “ATCG”.

(3) Dynamic permutation: Calculate the permutation rule value $R_{S}$ according to the pixel position value of the DNA-encoded matrix and perform the corresponding permutation to obtain the permutation image *S*. The expression of $R_{S}$ is
(4)$$R_{S}=\;\mathrm{mod}\;\left(i\left(1,1\right)+i\left(1,4N\right)+i\left(M,1\right)+i\left(M,4N\right),4\right),$$
where the DNA symbol is converted into binary according to “A = 00, G = 01, C = 10, T = 11”. Then, the addition operation of Equation (Equation 4) is performed, and mod(·) is the modulo operation.

Since the Modulo 4 operation is performed when calculating $R_{S}$, $R_{S}$ satisfies $R_{S}\in 0,1,2,3$. The matrix of $M\times N\left(M=4,N=4\right)$ illustrates the permutation according to $R_{S}$.

When $R_{S}=0$, transpose the symbol matrix *I*, as shown in Figure 2.

When $R_{S}=1$, from i=1 to $fix\left(M/2\right)$, the *i*-th and M+1−i-th of the symbol matrix *I* exchange the entire row, as shown in Figure 3.

When $R_{S}=2$, from i=2 to $fix\left(N/2\right)$, the *i*-th and $fix\left(N/2\right)+i-1$-th of the symbol matrix *I* exchange the entire column, as shown in Figure 4.

When $R_{S}=3$, from i=2 to $fix\left(M/2\right)$, the *i*-th and $fix\left(M/2\right)+i-1$-th of the symbol matrix *I* exchange the entire row, as shown in Figure 5.

(4) DNA decoding: Select the rules in Table 1 for dynamic DNA decoding of the permutation image *S* according to the corresponding values in sequence *B*, decode *S* in the raster scanning order, combine the four symbols into one pixel, and obtain a DNA-decoded matrix $E={\left\{e\left(i,j\right)\right\}}_{i=1.j=1}^{M,N}$ with *M* rows and *N* columns.

(5) Dynamic diffusion: According to the DNA-decoded matrix *E*, the chaotic matrix *X*, and the diffusion matrix *D*, dynamically calculate the diffusion rule value $R_{D}=r\left(i,j\right)\left(i=1,2,\ldots ,M;j=1,2,\ldots ,N\right)$ at different positions of *E*, where the mathematical expression of $r\left(i,j\right)$ is
(5)$$r\left(i,j\right)=\left\{\begin{array}{cc} \;\mathrm{mod}\;(e(M,N)⊕x(1,1),2) & \mathrm{if}\;i=1,j=1,\\ \;\mathrm{mod}\;(d(i-1,N)⊕x(i,1),2) & \mathrm{if}\;i\neq 1,j=1,\\ \;\mathrm{mod}\;(d(i,j-1)⊕x(i,j),2) & \mathrm{if}\;j\neq 1, \end{array}\right.$$
where $r\left(i,j\right)\in \left\{0,1\right\}$, mod(·) is the modulo operation, and ⊕ is the exclusive OR operation.

The diffusion rule value $R_{D}$ determines the diffusion rule performed by the pixel at different positions of *E*, and the diffusion image is $D={\left\{d\left(i,j\right)\right\}}_{i=1,j=1}^{M,N}$.

Figure 6 shows how to obtain the diffusion rule value $R_{D}=r\left(i,j\right)\left(i=1,2,\ldots ,M;j=1,2,\ldots ,N\right)$ for an image of 3×3.

The diffusion image $D={\left\{d\left(i,j\right)\right\}}_{i=1,j=1}^{M,N}$ is obtained as follows:(6)$$d\left(i,j\right)=\left\{\begin{array}{cc} F_{r\left(i,j\right)}\left(e(M,N),e(1,1),y(1,1)\right) & \mathrm{if}\;i=1,j=1,\\ F_{r\left(i,j\right)}\left(e(i,1),d(i-1,N),y(i,1)\right) & \mathrm{if}\;i\neq 1,j=1,\\ F_{r\left(i,j\right)}\left(e(i,j),d(i,j-1),y(i,j)\right) & \mathrm{if}\;j\neq 1, \end{array}\right.$$
where $F_{r\left(i,j\right)}\left(a,b,c\right)=\left\{\begin{array}{cc} a⊕b⊕c & \mathrm{if}\;r\left(i,j\right)=0,\\ \;\mathrm{mod}\;\left(a+b+c,256\right) & \mathrm{if}\;r\left(i,j\right)=1 \end{array},\right.$ and $a,b,c\in \left\{0,1,2\ldots ,255\right\}$. Figure 7 shows how to obtain the diffusion image $D={\left\{d\left(i,j\right)\right\}}_{i=1,j=1}^{3,3}$.

(6) Disconnect $K_{1}$, connect $K_{2}$ to position one, and use the diffusion image *D* as the input image for the next round of encryption.

(7) The second round of encryption: Repeat steps (3)–(6) to make $K_{2}$ connect to position two. Then, the final cipher image *C* can be obtained.

The decryption process is the inverse process of encryption, and decryption can be completed by operating the above steps in reverse order. The main steps of decryption are given here.

(1) Inverse dynamic diffusion: Record the inverse diffusion rule value as $R_{D}^{-1}=r^{-1}\left(i,j\right)\left(i=1,2,\ldots ,M;j=1,2,\ldots ,N\right)$. According to the inverse diffusion rule value $R_{D}^{-1}$, choose the inverse diffusion equation at different positions to obtain *E*,
(7)$$r^{-1}\left(i,j\right)=\left\{\begin{array}{cc} \;\mathrm{mod}\;(d(i,j-1)⊕x(i,j),2) & \mathrm{if}\;j\neq 1,\\ \;\mathrm{mod}\;(d(i-1,N)⊕x(i,1),2) & \mathrm{if}\;i\neq 1,j=1,\\ \;\mathrm{mod}\;(e(M,N)⊕x(1,1),2) & \mathrm{if}\;i=1,j=1. \end{array}\right.$$

The matrix $E={\left\{e\left(i,j\right)\right\}}_{i=1.j=1}^{M,N}$, before diffusion, is obtained as shown in Equation (Equation 7).
(8)$$e\left(i,j\right)=\left\{\begin{array}{cc} F_{r\left(i,j\right)}^{-1}\left(d(i,j),d(i,j-1),y(i,j)\right) & \mathrm{if}\;j\neq 1,\\ F_{r\left(i,j\right)}^{-1}\left(d(i,1),d(i-1,N),y(i,1)\right) & \mathrm{if}\;i\neq 1,j=1,\\ F_{r\left(i,j\right)}^{-1}\left(d(M,N),e(1,1),y(1,1)\right) & \mathrm{if}\;i=1,j=1, \end{array}\right.$$
where $F_{r\left(i,j\right)}^{-1}\left(a,b,c\right)=\left\{\begin{array}{cc} a⊕b⊕c & \mathrm{if}\;r\left(i,j\right)=0,\\ \;\mathrm{mod}\;\left(a+b+c,256\right) & \mathrm{if}\;r\left(i,j\right)=1, \end{array}\right.$ and $a,b,c\in \left\{0,1,2\ldots ,255\right\}$.

(2) Inverse DNA diffusion: The pixels in *E* are dynamically decoded according to the corresponding values in sequence *B* in the raster scanning order, and the corresponding four binary pairs are converted into the DNA symbol matrix *S* by the $b\left(4i-3\right),b\left(4i-2\right),b\left(4i-1\right),b\left(4i\right)$ column coding rules in Table 1.

(3) Inverse permutation: $R_{S}^{-1}$ is obtained to select the inverse permutation rule, and $R_{S}^{-1}=\;\mathrm{mod}\;\left(s\left(1,1\right)\dot{+}s\left(1,4N\right)\dot{+}s\left(M,1\right)\dot{+}s\left(M,4N\right),4\right)$.

(4) Inverse DNA encoding: According to the corresponding value in sequence *A*, the DNA coding rule in Table 1 is dynamically selected to reverse encode *I* to obtain the DNA-decoded matrix *P*.

During encryption, both the permutation rule value $R_{S}$ and the diffusion rule value $R_{D}$ will change with the number of rounds to achieve the purpose of dynamic rule selection, so as to flexibly use different permutation rules and diffusion rules. When the image is encrypted, different permutation rules will be implemented for different encryption rounds, so as to realize dynamic permutation at the image level. In dynamic diffusion, the diffusion rule value of each pixel will also change dynamically with the pixel position, and the diffusion rule value of each round will also change dynamically with the number of encryption rounds, achieving dynamic diffusion from the pixel level. For different images to be encrypted, $R_{S}$ depends on the special location pixels of the input image to be encrypted after DNA encoding and the different images to be encrypted by different permutation rule values. The diffusion rule value $R_{D}$ is mainly determined by the image before diffusion *E*. The permutation image *S* and the chaotic matrix *X*, and the diffusion rule value $R_{D}$ will also change as the number of rounds of encryption changes the DNA-decoded matrix *E* and the permutation image *S*.

Setting the number of feedback rounds to one can not only reflect the characteristics of dynamic diffusion rules changing with the number of rounds but also reflect the dynamic selection and the multi-parallel structure. Too many rounds will definitely affect the encryption efficiency. In this scheme, as the number of rounds changes, fewer rounds can be used to achieve the core advantages of the scheme, namely the dynamic, parallel structure and the dynamic diffusion rule matrix $R_{D}$. Through the global dynamic selection feature, the combination of encryption methods for any one-bit pair change is $2^{9}$, and the combination of encryption methods for any pixel change is $2^{27}$. This is the main difference between the parallel structure proposed in this paper and the existing non-parallel structure.

## 3. Security Analysis

### 3.1. Equivalent Key Analysis

The core of the scheme’s security lies in the ciphertext feedback mechanism and dynamic selection characteristics. Through these two characteristics, the cost of finding special plaintext pairs that are conducive to cracking is significantly higher. This section theoretically analyzes the ciphertext feedback mechanism and dynamic selection characteristics in this encryption scheme to illustrate the effect of the ciphertext feedback mechanism and the dynamic selection feature on the security of the scheme.

#### 3.1.1. Analysis of Ciphertext Feedback Mechanism in Diffusion

The expression of the ciphertext feedback mechanism reflected in the diffusion is shown in Equation (Equation 6). In order to better study the effect of the ciphertext feedback mechanism, let the image before diffusion be $E={\left\{e\left(i,j\right)\right\}}_{i=1,j=1}^{M,N}$, and the image after diffusion be $D={\left\{d\left(i,j\right)\right\}}_{i=1,j=1}^{M,N}$.

Transform Equation (Equation 6) into
(9)$$d\left(i,j\right)=\left\{\begin{array}{cc} O_{r(i,j)}\left(H\left(i,j\right),e\left(M,N\right)\right) & \;\mathrm{if}\;i=1,j=1,\\ O_{r(i,j)}\left(H\left(i,j\right),d\left(i-1,N\right)\right) & \;\mathrm{if}\;i\neq 1,j=1,\\ O_{r(i,j)}\left(H\left(i,j\right),d\left(i,j-1\right)\right) & \;\mathrm{if}\;j\neq 1, \end{array}\right.$$
where $H\left(i,j\right)=\left\{\begin{array}{cc} e(i,j)⊕y(i,j) & \;\mathrm{if}\;r(i,j)=0,\\ \;\mathrm{mod}\;(e(i,j)+y(i,j),256) & \;\mathrm{if}\;r(i,j)=1, \end{array}\right.$ is the diffusion-related factor, $O_{r(i,j)}\left(a,b\right)=\left\{\begin{array}{cc} a⊕b & \;\mathrm{if}\;r(i,j)=0,\\ \;\mathrm{mod}\;(a+b,256) & \;\mathrm{if}\;r(i,j)=1, \end{array}\right.$ and $a,b,c\in \left\{0,1,2\ldots ,255\right\}$.

**Proposition** **1.**
*If different images before diffusion $E^{\prime }={\left\{e^{\prime }\left(i,j\right)\right\}}_{i=1,j=1}^{M,N}$ and $E^{\prime \prime }={\left\{e^{\prime \prime }\left(i,j\right)\right\}}_{i=1,j=1}^{M,N}$ have $e^{\prime }\left(q,l\right)\neq e^{\prime \prime }\left(q,l\right)$ at $\left(q,l\right)$, then $\Delta H\left(q,l\right)=H^{\prime }\left(q,l\right)⊕H^{\prime \prime }\left(q,l\right)\neq 0$.*


**Proof.** For different images $E^{\prime }={\left\{e^{\prime }\left(i,j\right)\right\}}_{i=1,j=1}^{M,N}$ and $E^{\prime \prime }={\left\{e^{\prime \prime }\left(i,j\right)\right\}}_{i=1,j=1}^{M,N}$, there are the diffusion-related factors
$$H^{\prime }\left(i,j\right)=\left\{\begin{array}{cc} e^{\prime }\left(i,j\right)⊕y\left(i,j\right) & \;\mathrm{if}\;r(i,j)=0,\\ \;\mathrm{mod}\;\left(e^{\prime }\left(i,j\right)+y\left(i,j\right),256\right) & \;\mathrm{if}\;r(i,j)=1, \end{array}\right.$$
and
$$H^{\prime \prime }\left(i,j\right)=\left\{\begin{array}{cc} e^{\prime \prime }\left(i,j\right)⊕y\left(i,j\right) & \;\mathrm{if}\;r(i,j)=0,\\ \;\mathrm{mod}\;\left(e^{\prime \prime }\left(i,j\right)+y\left(i,j\right),256\right) & \;\mathrm{if}\;r(i,j)=1. \end{array}\right.$$
Respectively, the corresponding images are $D^{\prime }={\left\{d^{\prime }\left(i,j\right)\right\}}_{i=1,j=1}^{M,N}$ and $D^{\prime \prime }={\left\{d^{\prime \prime }\left(i,j\right)\right\}}_{i=1,j=1}^{M,N}$.The diffusion-related factor at $\left(q,l\right)$ is expressed as $H^{\prime }\left(q,l\right)=e^{\prime }\left(q,l\right)⊕y\left(q,l\right)$ and $H^{\prime \prime }\left(q,l\right)=e^{\prime \prime }\left(q,l\right)⊕y\left(q,l\right)$. Then, $\Delta H\left(q,l\right)=H^{\prime }\left(q,l\right)⊕H^{\prime \prime }\left(q,l\right)=e^{\prime }\left(q,l\right)⊕e^{\prime \prime }\left(q,l\right)\neq 0$. Proposition 1 is proved.  □

From Proposition 1, $\Delta H\left(q,l\right)=H^{\prime }\left(q,l\right)⊕H^{\prime \prime }\left(q,l\right)\neq 0$, where $q\in \left\{i=1,2,\ldots ,M\right\}$ and $l\in \left\{j=1,2,\ldots ,N\right\}$. According to Equation (Equation 9), it is found that the difference $\Delta H\left(q,l\right)$ will be passed to the next pixel after the ciphertext feedback mechanism, making the value of $\Delta d\left(i,j\right)=d^{\prime }\left(i,j\right)⊕d^{\prime \prime }\left(i,j\right)\left(i=q,q+1,\ldots ,M;j=l,l+1,\ldots ,N\right)$ unpredictable.

Supposing that the diffusion images in the first round are $D_{1}^{\prime }=d_{1}^{\prime }\left(i,j\right)$ and $D_{1}^{\prime \prime }=d_{1}^{\prime \prime }\left(i,j\right)$, the diffusion images in the second round are $D_{2}^{\prime }=d_{2}^{\prime }\left(i,j\right)$ and $D_{2}^{\prime \prime }=d_{2}^{\prime \prime }\left(i,j\right)$, and the final cipher images are $C^{\prime }=c^{\prime }\left(i,j\right)=D_{2}^{\prime }$ and $C^{\prime \prime }=c^{\prime \prime }\left(i,j\right)=D_{2}^{\prime \prime }$, respectively.

For the first round of encryption, $\Delta d_{1}\left(i,j\right)=d_{1}^{\prime }\left(i,j\right)⊕d_{1}^{\prime \prime }\left(i,j\right)$ is unpredictable at (i=q,q+1,…,M;j=l,l+1,…,N), but in the second round of encryption, $d_{1}^{\prime }\left(i,j\right)$ and $d_{1}^{\prime \prime }\left(i,j\right)$ are used as the input images, and the unpredictability of $d_{1}^{\prime }\left(i,j\right)$ and $d_{1}^{\prime \prime }\left(i,j\right)$ is transmitted to other positions of the image by DNA encoding, dynamic permutation, and DNA decoding, applying the unpredictability of a single pixel to all pixels in the image. In addition, the input images $D_{1}^{\prime }$ and $D_{1}^{\prime \prime }$ in the second round are uncontrollable for the attacker, and it is difficult to directly select a special plaintext pair to obtain a partially controllable $\Delta d\left(q,l\right)=c^{\prime }\left(q,l\right)⊕c^{\prime \prime }\left(q,l\right)=d_{2}^{\prime }\left(q,l\right)⊕d_{2}^{\prime \prime }\left(q,l\right)\neq 0$ by chosen-plaintext attack, so that the equivalent key $Y={\left\{y\left(i,j\right)\right\}}_{i=1,j=1}^{M,N}$ cannot be cracked.

#### 3.1.2. Diffusion Rule Value Difference Analysis $\Delta R_{D}=r\left(i,j\right)\left(i=1,2,\ldots ,M;j=1,2,\ldots ,N\right)$

The image after the first round of DNA decoding is $E_{1}={\left\{e_{1}\left(i,j\right)\right\}}_{i=1,j=1}^{M,N}$. The image after the second round of DNA decoding is $E_{2}={\left\{e_{2}\left(i,j\right)\right\}}_{i=1,j=1}^{M,N}$. The image after the first round of dynamic diffusion is $D_{1}={\left\{d_{1}\left(i,j\right)\right\}}_{i=1,j=1}^{M,N}$, and the image after the second round of dynamic diffusion is the final cipher image $C={\left\{c(i,j)\right\}}_{i=1,j=1}^{M,N}=D_{2}={\left\{d_{2}\left(i,j\right)\right\}}_{i=1,j=1}^{M,N}$.

For the ciphertext feedback mechanism of the first round of the diffusion rule $\Delta R_{D1}=r\left(i,j\right)\left(i=1,2,\ldots ,M;j=1,2,\ldots ,N\right)$, the expression is as follows:(10)$$r_{1}\left(i,j\right)=\left\{\begin{array}{cc} \;\mathrm{mod}\;(e_{1}\left(M,N\right)⊕x\left(1,1\right),2) & \;\mathrm{if}\;i=1,j=1,\\ \;\mathrm{mod}\;(d_{1}\left(i-1,N\right)⊕x\left(i,1\right),2) & \;\mathrm{if}\;i\neq 1,j=1,\\ \;\mathrm{mod}\;(d_{1}\left(i,j-1\right)⊕x\left(i,j\right),2) & \;\mathrm{if}\;j\neq 1. \end{array}\right.$$

The diffusion rule value difference $\Delta R_{D1}=r_{1}\left(i,j\right)\left(i=1,2,\ldots ,M;j=1,2,\ldots ,N\right)$ can be simplified to $r_{1}\left(i,j\right)=\;\mathrm{mod}\;\left(G_{1}\left(i,j\right)⊕x\left(i,j\right),2\right)$, where
$$G\left(i,j\right)=\left\{\begin{array}{cc} e(M,N) & \;\mathrm{if}\;i=1,j=1,\\ d(i-1,N) & \;\mathrm{if}\;i\neq 1,j=1,\\ d(i,j-1) & \;\mathrm{if}\;j\neq 1, \end{array}\right.$$
is the extracted ciphertext-related factor.

To better study the effect of ciphertext feedback mechanisms in this process, it is assumed that there are different
$$G_{1}^{\prime }\left(i,j\right)=\left\{\begin{array}{cc} e_{1}^{\prime }\left(M,N\right) & \;\mathrm{if}\;i=1,j=1,\\ d^{\prime }\left(i-1,N\right) & \;\mathrm{if}\;i\neq 1,j=1,\\ d^{\prime }\left(i,j-1\right) & \;\mathrm{if}\;j\neq 1, \end{array}\right.$$
and
$$G_{1}^{\prime \prime }\left(i,j\right)=\left\{\begin{array}{cc} e_{1}^{\prime \prime }\left(M,N\right) & \;\mathrm{if}\;i=1,j=1,\\ d^{\prime \prime }\left(i-1,N\right) & \;\mathrm{if}\;i\neq 1,j=1,\\ d^{\prime \prime }\left(i,j-1\right) & \;\mathrm{if}\;j\neq 1, \end{array}\right.$$
where the subscript 1 represents the first round of encryption, i.e., $\Delta G_{1}=G_{1}^{\prime }⊕G_{1}^{\prime \prime }\neq 0$. $G_{1}^{\prime }\left(i,j\right)$. The diffusion rules corresponding to $G_{1}^{\prime }\left(i,j\right)$ and $G_{1}^{\prime \prime }\left(i,j\right)$ are $R_{D1}^{\prime }=r_{1}^{\prime }\left(i,j\right)=\;\mathrm{mod}\;\left(G_{1}^{\prime }\left(i,j\right)⊕x\left(i,j\right),2\right)$ and $R_{D1}^{\prime \prime }=r_{1}^{\prime \prime }\left(i,j\right)=\;\mathrm{mod}\;\left(G_{1}^{\prime \prime }\left(i,j\right)⊕x\left(i,j\right),2\right)$, respectively, where there must be $G_{1}^{\prime }\left(i,j\right)⊕x\left(i,j\right)\neq G_{1}^{\prime \prime }\left(i,j\right)⊕x\left(i,j\right)$.

Specifically, suppose $G_{1}^{\prime }\left(i,j\right)$ and $G_{1}^{\prime \prime }\left(i,j\right)$ exist, and $\Delta G_{1}\left(q,l\right)=G_{1}^{\prime }\left(q,l\right)⊕G_{1}^{\prime \prime }\left(q,l\right)\neq 0$, where q∈{0,1,2,…,M} and l∈{0,1,2,…,N}. After the ciphertext feedback mechanism, the unpredictability of (q,l) is passed to the next pixel of (q,l), and so on, eventually making the value of $\Delta G_{1}\left(i,j\right)\left(M\geq i\geq q,N\geq j\geq l\right)$ unpredictable, meaning that $G_{1}^{\prime }\left(i,j\right)⊕x\left(i,j\right)$ and $G_{1}^{\prime \prime }\left(i,j\right)⊕x\left(i,j\right)\left(M\geq i\geq q,N\geq j\geq l\right)$ in the first round of diffusion rule expressions are unpredictable at (M≥i≥q,N≥j≥l). For $R_{D1}^{\prime }=r_{1}^{\prime }\left(i,j\right)=\;\mathrm{mod}\;\left(G_{1}^{\prime }\left(i,j\right)⊕x\left(i,j\right),2\right)$ and $R_{D1}^{\prime }=r_{1}^{\prime }\left(i,j\right)=\;\mathrm{mod}\;\left(G_{1}^{\prime }\left(i,j\right)⊕x\left(i,j\right),2\right)$, since $R_{D1}^{\prime }$ and $R_{D1}^{\prime \prime }$ in $G_{1}^{\prime }\left(i,j\right)⊕x\left(i,j\right)$ and $G_{1}^{\prime \prime }\left(i,j\right)⊕x\left(i,j\right)$ are unmeasurable at (M≥i≥q,N≥j≥l), the number of diffusion rules that need to be exhausted are in the range of (M≥i≥q,N≥j≥l) is $2^{\left(M-q)(M-l\right)}$.

In the second round of encryption, the ciphertext-related factors in the corresponding diffusion rules of $C^{\prime }$ and $C^{\prime \prime }$ are
$$G_{2}^{\prime }\left(i,j\right)=\left\{\begin{array}{cc} e_{2}^{\prime }\left(M,N\right) & \;\mathrm{if}\;i=1,j=1,\\ c^{\prime }\left(i-1,N\right) & \;\mathrm{if}\;i\neq 1,j=1,\\ c^{\prime }\left(i,j-1\right) & \;\mathrm{if}\;j\neq 1, \end{array}\right.$$
and
$$G_{2}^{\prime \prime }\left(i,j\right)=\left\{\begin{array}{cc} e_{2}^{\prime \prime }\left(M,N\right) & \;\mathrm{if}\;i=1,j=1,\\ c^{\prime \prime }\left(i-1,N\right) & \;\mathrm{if}\;i\neq 1,j=1,\\ c^{\prime \prime }\left(i,j-1\right) & \;\mathrm{if}\;j\neq 1, \end{array}\right.$$
respectively. Since the unpredictability of $C^{\prime }$ and $C^{\prime \prime }$ also makes $G_{2}^{\prime }\left(i,j\right)$ and $G_{2}^{\prime \prime }\left(i,j\right)$ unpredictable, this makes the second round of the permutation rule $R_{D2}=r\left(i,j\right)\left(i=1,2,\ldots ,M;j=1,2,\ldots ,N\right)$ more unpredictable. Based on this, and because of the dynamic nature of r(i,j), r(i,j) in each location is unpredictable, the attacker needs to exhaust all cases. For a M×N-size image, the second round of the diffusion rule value $R_{D2}$ needs to be exhausted $2^{M\times N}$, and the total number of exhaustive times for $R_{D1}$ and $R_{D2}$ in the case of two rounds of encryption is $2^{M\times N}+2^{\left(M-q)(M-l\right)}\geq 2^{M\times N}$.

According to the development of the limit of exhaustive attacks based on Moore’s Law, the limit of exhaustive attacks in 2022 is $2^{87}$, and the limit of exhaustive attacks in 2050 will be $2^{109}$ [39]. For the existing effective image size, it is easy to satisfy that $2^{M\times N}$ is larger than $2^{109}$, and there are 256×256, 512×512, and 1024×1024, which are far greater than the $2^{109}$ required for the key space.

### 3.2. Key Space Analysis

Any chaotic image encryption scheme has a key space larger than $2^{100}$ to ensure that it can withstand brute force attacks. The key space mentioned here means that the chaotic digital image system uses a key with a specified length.

In this scheme, the 2D-LSM system is in a chaotic state within the parameter range from h(i),w(i)∈[0,1] to α,β∈[1,100]. With a finite precision of $10^{-15}$, there are $S_{h}=10^{15}$, $S_{w}=10^{15}$, $S_{\alpha }=9.9\times 10^{16}$, and $S_{\beta }=9.9\times 10^{16}$, and the calculation formula for a set of parameter key spaces is as follows:(11)$$S_{h}\times S_{w}\times S_{\alpha }\times S_{\beta }=9.801\times 10^{63}.$$

Because two sets of key parameters are set, the overall key space of this scheme is $S={\left(9.801\times 10^{63}\right)}^{2}=9.61\times 10^{129}\approx 2^{421}$, which is much larger than $2^{100}$, which meets the key space requirements of the encryption scheme.

## 4. Simulation Experiments and Performance Analysis

The experimental hardware platform is a PC, and the processor is Ryzen 5 5600 G AMD, The benchmark frequency is 3.90 GHz, the memory size is 16G, the hard disk is a 128G SSD, and the HDD is 1 T. The software environment is the Windows 10 operating system and Matlab R2019a.

In this section, to demonstrate the security of our scheme, a grayscale image of the size 256×256 is used as the plain image. The initial keys are

$h_{1}\left(0\right)=0.2333$, $w_{1}\left(0\right)=0.25$, $\alpha_{1}=3$, $\beta_{1}=4$, $h_{2}\left(0\right)=0.28$, $w_{2}\left(0\right)=0.289$, $\alpha_{2}=4$,

$\beta_{2}=3$.

The original plain image encrypted with the number of feedback rounds is one to illutrate the encryption performance indicators. The relevant experimental results are shown in Figure 8, where (a), (d), and (g) are the plain images of Lena, Peppers, and Cameraman, respectively; (b), (e), and (h) are the cipher images of Lena, Peppers, and Cameraman, respectively; and (c), (f), and (i) are the decrypted images of Lena, Peppers, and Cameraman, respectively.

### 4.1. Histogram Analysis

An image histogram is a frequency statistic for each grayscale level in an image. The histogram shows the distribution of grayscale in the image. For the distribution of pixel intensity in the image, the histogram of the cipher image obtained by a secure encryption scheme should be as flat and uniform as possible. A more evenly distributed histogram means a better ability to resist statistical attacks, as shown in Figure 9. The plain images of Lena, Cameraman, and Peppers with their histograms and the ciphertexts with their corresponding histograms are shown in Figure 9. It can be seen that the grayscale distribution of their original image has many peaks and valleys, but, in the encrypted grayscale image, the grayscale distribution is very uniform. Therefore, it can be confirmed that the scheme we designed has the performance of resisting statistical attacks.

### 4.2. Correlation Analysis

Since adjacent pixels of common images are highly correlated in horizontal, vertical, and diagonal directions, this indicates that adjacent pixels often have similar and predictable features. An ideal image encryption scheme should have sufficiently low correlation coefficients in the horizontal, vertical, and diagonal directions to resist statistical attacks.

To highlight the influence of the encryption scheme proposed in this paper on eliminating the high correlation of planar images, their correlation in the horizontal, vertical and diagonal directions was calculated by
(12)$$r_{u,v}=\frac{cov(u,v)}{\sqrt{D(u)D(v)}}.$$
where
(13)$$\left\{\begin{array}{c} cov\left(u,v\right)\;=\frac{1}{N}\sum_{i=1}^{N}\left(u_{i}-E\left(u\right)\right)\left(v_{i}-E\left(v\right)\right),\\ D\left(u\right)=\frac{1}{N}\sum_{i=1}^{N}{\left(u_{i}-E\left(u\right)\right)}^{2},\\ E\left(u\right)=\frac{1}{N}\sum_{i=1}^{N}u_{i}. \end{array}\right.$$
and *N* is the number of randomly chosen adjacent pixel pairs along the horizontal direction, vertical direction, and diagonal direction in both the plain image and its cipher image. $u_{i}$ and $v_{i}$ are the *i*-th items of the two adjacent pixel sequences *u* and *v*.

Figure 10 demonstrates the adjacent pixel correlation plots of the arbitrarily chosen 2000 sets of nearby pixels in Lena, Cameraman, and Peppers along the horizontal, vertical, and diagonal orders. It can be observed that, in each figure, the X-axis indicates the three images, while the Y-Z plane plots the values of the adjacent pixels. The adjacent pixel pairs of the plain images are mostly on or close to the diagonal lines, indicating that these adjacent pixels exhibit strong correlations. However, the adjacent pixel pairs for all cipher images are distributed quite randomly across the Y-Z phase plane, demonstrating that they exhibit weak correlations. This indicates that our proposed scheme can efficiently decorrelate the high correlations of the plain images.

The correlation distribution results are shown in Table 2, where we calculate the correlation coefficients of adjacent pixels in the horizontal, vertical, and diagonal directions of Lena, Cameraman, and Peppers and images 4.2.05, 4.2.06, and 4.2.07 with 512×512, and it can be clearly seen that the correlation coefficient of the original image is close to 1, while the correlation coefficient of the encrypted image is close to 0 in all directions.

### 4.3. NPCR and UACI Tests

A differential attack is a common security attack model. In the broadest sense, it refers to an attack for tracing how differences in information input can affect the resultant difference at the output and exploiting such properties to recover the secret key (cryptography key). An image encryption scheme exhibits high performance in resisting differential attacks if it possesses the characteristics of diffusion and the avalanche effect. The above characteristics indicate that a slight change in the plaintexts can spread over all of the data in the ciphertexts.

Therefore, the number of pixel change rates (*NPCR*) and a unified average changing intensity (*UACI*) are proposed to better measure the diffusion and avalanche effect characteristics in an encryption scheme. Security (resistance to differential attacks) is associated with high *UACI*/*NPCR* values. The calculation formula is as follows
(14)$$NPCR\left(T_{1},T_{2}\right)=\frac{1}{M\times N}\sum_{i=1}^{M}\sum_{j=1}^{N}\left|Sign\left(t_{1}\left(i,j\right)-t_{2}\left(i,j\right)\right)\right|\times 100\%,$$
(15)$$UACI\left(T_{1},T_{2}\right)=\frac{1}{M\times N}\sum_{i=1}^{M}\sum_{j=1}^{N}\frac{\left|t_{1}\left(i,j\right)-t_{2}\left(i,j\right)\right|}{255-0}\times 100\%,$$
where two images of the same size are denoted as $T_{1}$ and $T_{2}$, the size of the image is L=M×N, $T_{1}\left(i,j\right)$ and $T_{2}\left(i,j\right)$ are the pixel values of the corresponding coordinates (i,j) in the images, and Sign(·) is the sign function as
(16)$$Sign\left(x\right)=\left\{\begin{array}{cc} 1 & \;\mathrm{if}\;x>0,\\ 0 & \;\mathrm{if}\;x=0,\\ -1 & \;\mathrm{if}\;x<0. \end{array}\right.$$

Given the significance level λ, the critical *NPCR* score $N_{\lambda }^{*}$ is obtained as
(17)$$N_{\lambda }^{*}=\frac{G-\phi^{-1}\left(\lambda \right)\sqrt{G/L}}{G+1}.$$
where *G* indicates the largest allowed pixel value, and $\phi^{-1}\left(\lambda \right)$ is the inverse cumulative density function of the standard normal distribution N(0,1). The critical UACI scores $\left(U_{\lambda }^{*-},U_{\lambda }^{*+}\right)$ with the given λ can be obtained using
(18)$$\left\{\begin{array}{c} U_{\lambda }^{*-}=\mu_{u}-\phi^{-1}\left(\lambda /2\right)\times \sigma_{u},\\ U_{\lambda }^{*+}=\mu_{u}+\phi^{-1}\left(\lambda /2\right)\times \sigma_{u}, \end{array}\right.$$
where $\mu_{u}=\frac{G+2}{3G+3}$ and $\sigma_{u}^{2}=\frac{\left(G+2\right)\left(G^{2}+2G+3\right)}{18L\times G{\left(G+1\right)}^{2}}$.

An encryption algorithm can pass the test if the calculated *UACI* value is within the range $\left(U_{\lambda }^{*-},U_{\lambda }^{*+}\right)$.

Using Equation (Equation 19) to calculate the median value of the confidence interval, compare the average value of *UACI* calculated by different images of the the same size. The closer to ${\bar{U}}_{\lambda }^{*}$, the more stable the *UACI* is
(19)$${\bar{U}}_{\lambda }^{*}=\frac{\left(U_{\lambda }^{*-}+U_{\lambda }^{*+}\right)}{2}.$$

In Table 3, the *NPCR* and *UACI* values of multiple encrypted images of different sizes in our scheme are compared with those in other schemes. The numbers in bold indicate the best indicators in the comparison scheme, and the numbers with underlines indicate that they failed the test. We find that the average *NPCR* and *UACI* values of our encrypted 256×256 images are 99.6084 and 33.4645, respectively. The *NPCR* value of reference [38] is 99.5818, which is the closest to the theoretical value, but its pass rate in the *NPCR* index is only 5/6. By giving priority to the pass rate, our *NPCR* value is closest to the theoretical value 99.5693 of the 256×256 image in Table 3, and the corresponding *UACI* value is also closest to the median value of the confidence interval ${\bar{U}}_{\lambda }^{*}=33.46355$.

The average values of *NPCR* and *UACI* calculated by different images of 512×512 are 99.6063 and 33.4653. Although the *NPCR* value in reference [38] is 99.5818, which is closest to the theoretical value, its pass rate in the *NPCR* index is only 16/18. By giving priority to the pass rate, our test value is closer to the theoretical *NPCR* value of 99.5893 for this size, and the corresponding *UACI* is also closest to the median of the confidence interval ${\bar{U}}_{\lambda }^{*}=33.46355$.

The average values of *NPCR* and *UACI* calculated by different images of 1024×1024 are 99.6063 and 33.4602. Although the *NPCR* value in reference [38] is 99.6037, which the closest to the theoretical value, its pass rate in the *NPCR* index is only 3/4. By giving priority to the pass rate, our test value is closer to the theoretical *NPCR* value of 99.5994 for this size, and the corresponding *UACI* is also closest to the median of the confidence interval ${\bar{U}}_{\lambda }^{*}=33.46355$.

In summary, our scheme has a high pass rate for *NPCR* and *UACI* indicators when encrypting images of different sizes, and the average values of *NPCR* and *UACI* obtained under different sizes of images are closer to the theoretical values. It shows that our scheme has a strong ability to resist differential attacks. Therefore, it can be verified that this scheme can resist differential attacks, and it also has certain advantages compared to other schemes.

### 4.4. Global Shannon Entropy and Local Local Shannon Entropy

Global Shannon entropy is an important indicator that reflects the random characteristics of image information. It is generally believed that the larger the global Shannon entropy, the stronger the uncertainty of the image (the greater the amount of information) and the less visible information. It is used to measure the distribution of image pixels. Their global Shannon entropy can be calculated as
(20)$$H=-\sum_{i=1}^{G}p\left(i\right)\log_{2}p\left(i\right),$$
where *G* indicates the largest allowed pixel value, and p(i) represents the probability of the occurrence of the pixel value *i*.

The theoretical value of the global Shannon entropy *H* intended for an eight-bit grayscale random image is nearer to eight. Here, the images with sizes of 256×256, 512×512, and 1024×1024 are selected, and the results are shown in Table 4.

For an image of the size 256×256, the best rates of [14,25] are 2/6 and 1/6; the best rate of our proposed scheme is 3/6. For an image of the size 512×512, the best rates of [14,25] are 1/18 and 11/18; the best rate of our proposed scheme is 13/18. For an image of the size 1024×1024, the best rates of [14,25] are 2/4 and 4/4; the best rate of our proposed scheme is 4/4. In total, [14] has a best rate of 5/28, [25] has a best rate of 16/28, and our scheme has a best rate of 21/28. The test results show that our proposed scheme has a better performance in the global Shannon entropy test, and the best rate is relatively good.

Local Shannon entropy is an important indicator to reflect the randomness of local regions [13]. It is generally believed that the confidence interval of a local Shannon entropy is [7.9019014,7.9030373]. The local Shannon entropy in this interval indicates that the image shows strong randomness in the local area.

Here, we define the local Shannon entropy measure for 30 local image blocks with 1936 pixels as
(21)$$\bar{H_{30,1936}}\left(S\right)=\sum_{i=1}^{30}\frac{H\left(S_{i}\right)}{30},$$
where $S_{i}$ is one of the randomly select non-overlapping image blocks with 1936 pixels within the image *S*. $H\left(S_{i}\right)\left(i=1,2,\cdots ,30\right)$ is computed by Shannon entropy via Equation (Equation 20).

The image sizes images are 256×256, 512×512, and 1024×1024, respectively, and the results are shown in Table 5.

It can be seen that the pass rates of [5,25] are both 2/6, and the pass rate of our proposed scheme is 5/6 for an image of the size 256×256. The pass rates of [5,25] are 13/18 and 18/18, and the pass rate of our proposed scheme is 17/18 for an image of the size 512×512. The pass rates of [5,25] are 3/4 and 4/4, and the pass rate of our proposed scheme is 4/4 for an image of the size 256×256. In total, ref. [5] has a pass rate of 20/28, ref. [25] has a pass rate of 24/28, and our scheme has a pass rate of 26/28.The comparison results show that our proposed scheme has a better overall performance in the local information entropy test, a relatively better pass rate, higher randomness, less visible information, and a better encryption performance.

### 4.5. Sensitivity Analysis

A cryptographic system with a good security performance must be key-sensitive, that is, a small change in the key will cause significant differences between the encrypted images and the decrypted images. Modify only minor changes to $\beta_{1}=4+10^{-15}$ for key susceptibility testing.

During the encryption, the Lena image is encrypted using the original key and a slightly changed key, respectively. The original Lena image is shown in Figure 11a, the cipher image with the original key is shown in Figure 11b, the cipher image with a slightly changed key is shown in Figure 11c, and the difference between the two cipher images is shown in Figure 11d. It indicates that a slight change in the plain image can spread over all of the data in the cipher images.

During the decryption, the same cipher image of Lena is decrypted with the correct key and with a slightly changed key, respectively. The original image is featured in Figure 12a, the encrypted image with the original key is featured in Figure 12b, the decrypted image with a slightly changed key is featured in Figure 12c, and the image decrypted by the original key is featured in Figure 12d.

## 5. Conclusions

We propose an image chaos encryption scheme based on global dynamic selection, the main work of which includes the following aspects:1.Design a multi-parallel structure to achieve dynamic selection.2.Dynamic selection of DNA encoding rules using chaotic sequences.3.Calculate the permutation rule according to the pixel position value of the DNA-encoded matrix and perform the corresponding permutation to obtain the permutation image.4.The diffusion rule obtained by the ciphertext feedback mechanism is introduced to determine the dynamic diffusion performed, and the image after the diffusion is obtained.

Compared with the existing local dynamic selection, the main advantage of this scheme is that it can realize global dynamic selection. According to the results of Lenstra et al., under the condition of limited years, if the cracking difficulty of this scheme is greater than that of an exhaustive attack, it has no attack value, so the equivalent key cannot be cracked. A theoretical analysis and a numerical analysis verify the feasibility of the scheme.

## Figures and Tables

**Figure 1 entropy-25-00476-f001:**
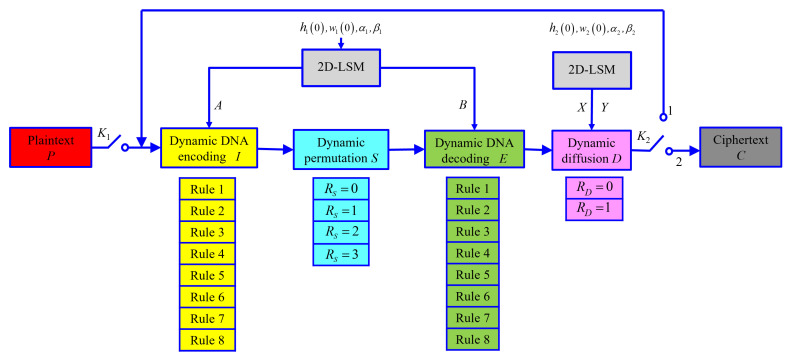
Block diagram of global dynamic encryption.

**Figure 2 entropy-25-00476-f002:**
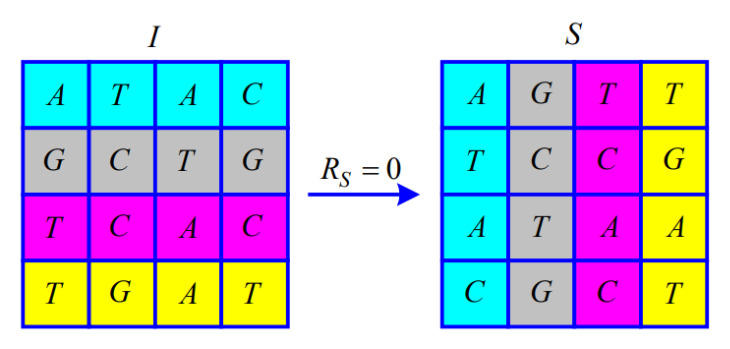
Illustration of the permutation for $R_{S}=0$.

**Figure 3 entropy-25-00476-f003:**
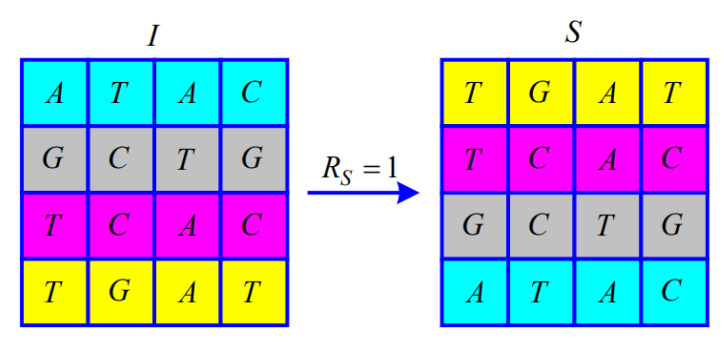
Illustration of the permutation for $R_{S}=1$.

**Figure 4 entropy-25-00476-f004:**
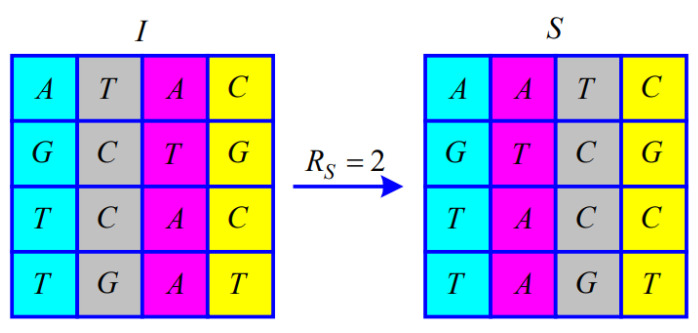
Illustration of the permutation for $R_{S}=2$.

**Figure 5 entropy-25-00476-f005:**
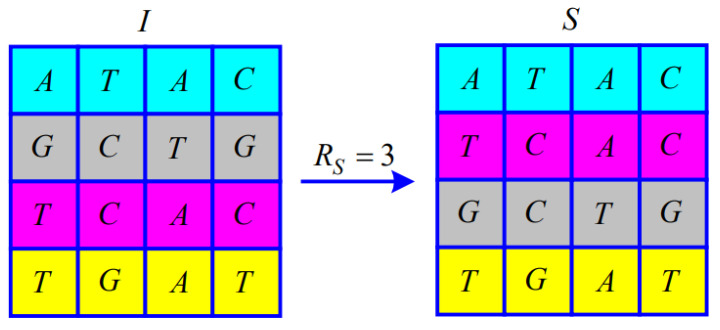
Illustration of the permutation for $R_{S}=3$.

**Figure 6 entropy-25-00476-f006:**
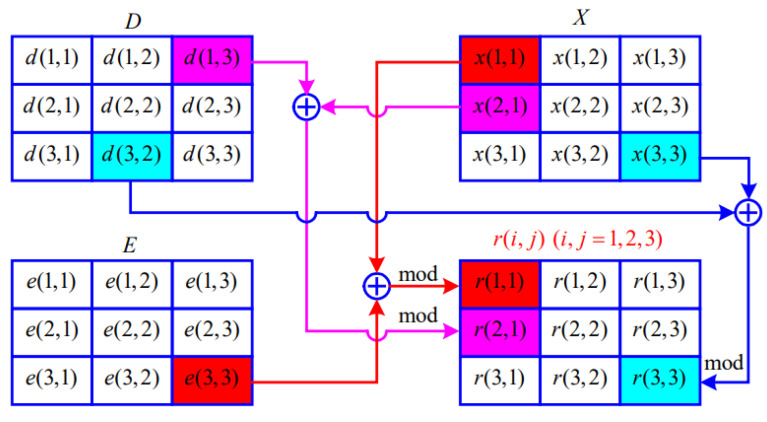
A schematic diagram of acquisition of diffusion rules.

**Figure 7 entropy-25-00476-f007:**
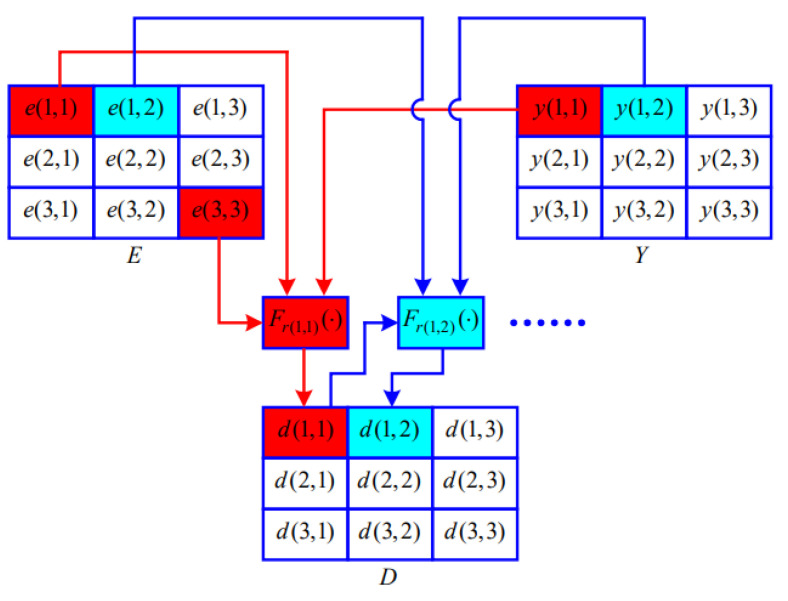
A schematic diagram of acquisition of *D*.

**Figure 8 entropy-25-00476-f008:**
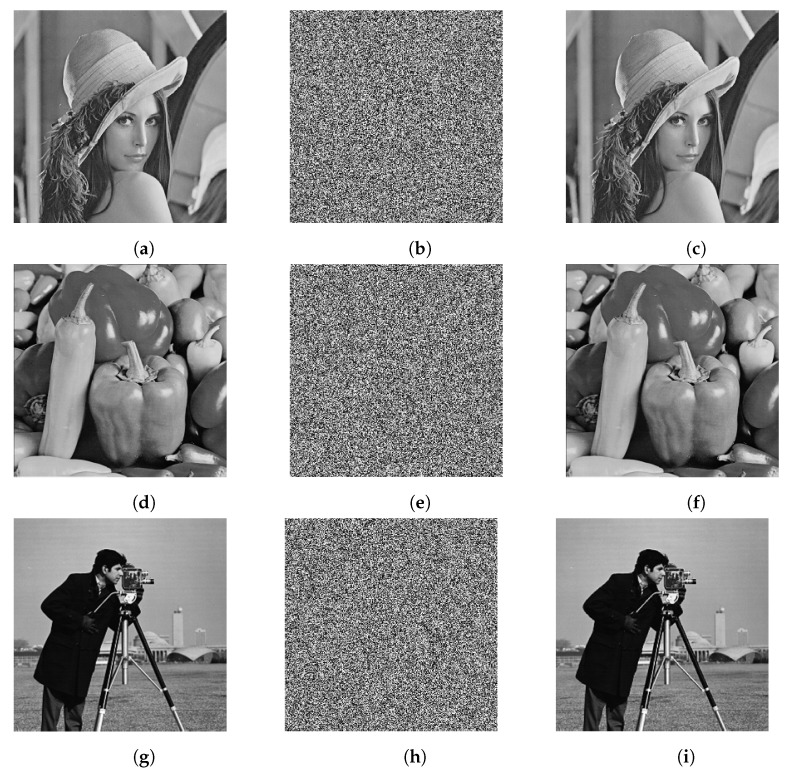
Experimental results: (**a**) Lena original image, (**b**) Lena encrypted image, (**c**) Lena decrypted image, (**d**) Peppers original image, (**e**) Peppers encrypted image, (**f**) Peppers decrypted image, (**g**) Cameraman original image, (**h**) Cameraman encrypted image, (**i**) Cameraman decrypted image.

**Figure 9 entropy-25-00476-f009:**
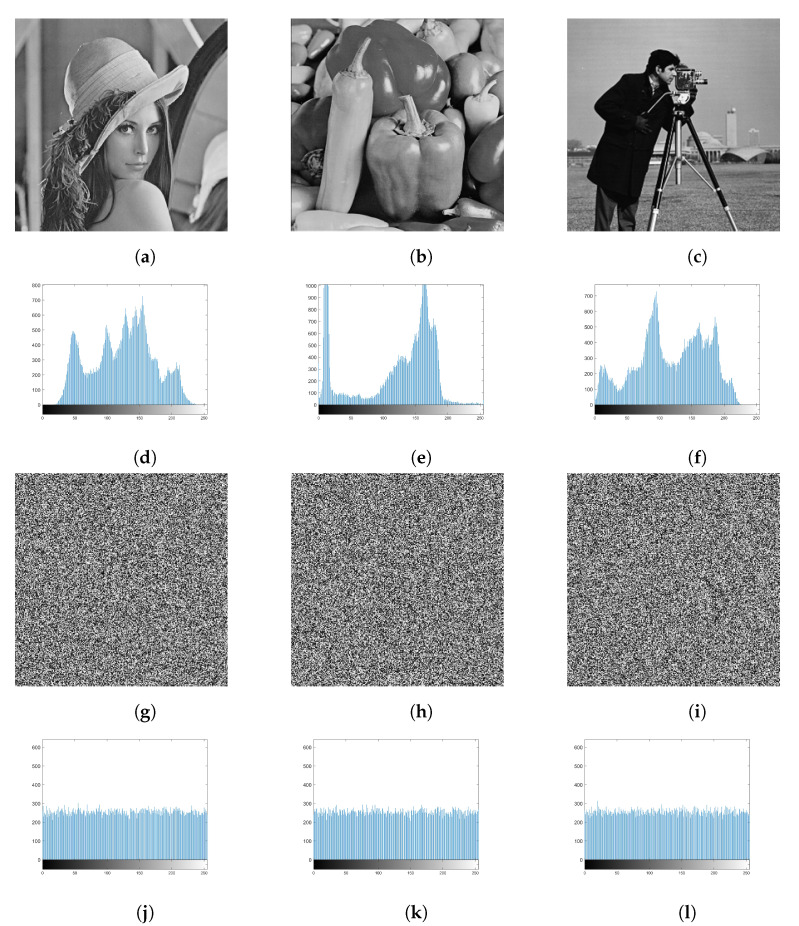
Experimental result: (**a**) plain image of Lena, (**b**) plain image of Peppers, (**c**) plain image of Cameraman, (**d**) histogram of Lena plain image, (**e**) histogram of a Peppers plain image, (**f**) histogram of Cameraman plain image, (**g**) cipher image of Lena, (**h**) cipher image of Peppers, (**i**) cipher image of Cameraman, (**j**) histogram of Lena’s cipher image, (**k**) histogram of Peppers’ cipher image, (**l**) histogram of Cameraman’s cipher image.

**Figure 10 entropy-25-00476-f010:**
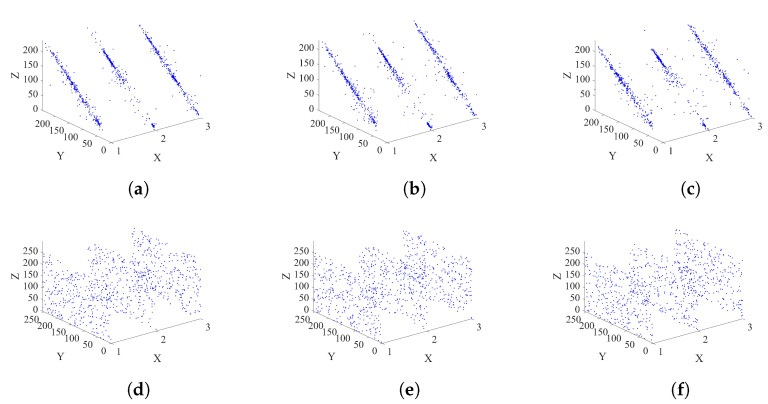
Histograms of Lena, Cameraman, and Peppers: (**a**) horizontal adjacent pixel pairs of three plain images, (**b**) vertical adjacent pixel pairs of three plain images, (**c**) diagonal adjacent pixel pairs of three plain images, (**d**) horizontal adjacent pixel pairs of three cipher images, (**e**) vertical adjacent pixel pairs of three cipher images, (**f**) diagonal adjacent pixel pairs of three cipher images. In each figure, the X-axis denotes the index of the three images, while the Y-Z plane plots the pixel pairs.

**Figure 11 entropy-25-00476-f011:**
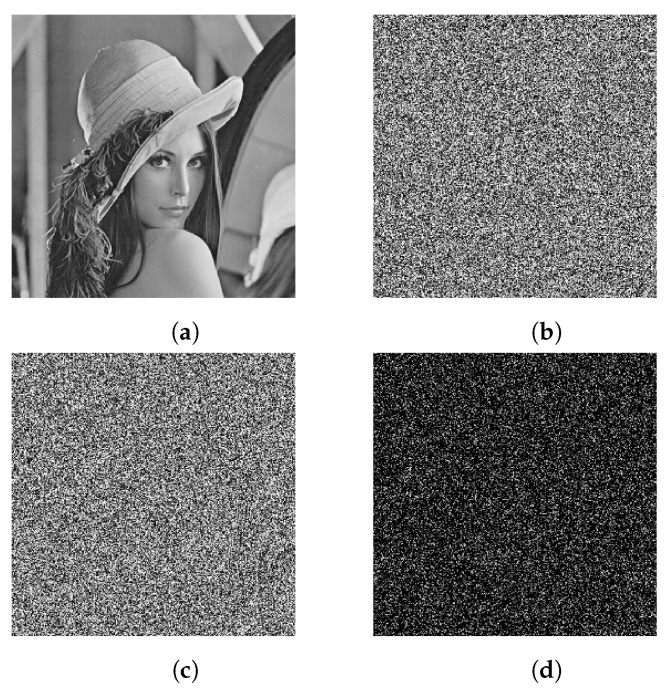
Key sensitivity test for image encryption: (**a**) plain image of Lena, (**b**) encrypted image with the original key, (**c**) encrypted image with a slightly changed key, (**d**) cipher image with horizontal adjacent pixel distribution.

**Figure 12 entropy-25-00476-f012:**
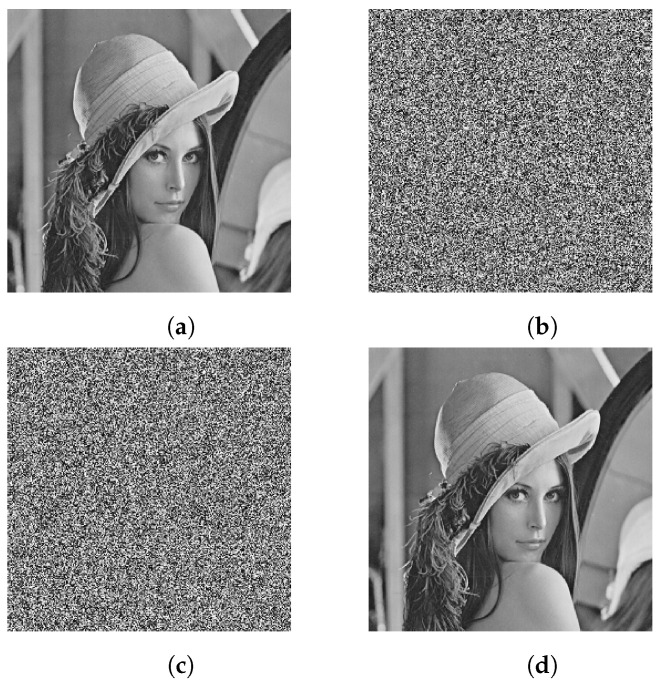
Key sensitivity test for image decryption: (**a**) plain image of Lena, (**b**) encrypted image with original key, (**c**) decrypted image with a slightly changed key, (**d**) decrypted image with the correct key.

**Table 1 entropy-25-00476-t001:** Eight kinds of DNA coding rules.

1	2	3	4	5	6	7	8
00-A	00-A	00-C	00-C	00-G	00-G	00-T	00-T
01-C	01-G	01-A	01-T	01-A	01-T	01-C	01-G
10-G	10-C	10-T	10-A	10-T	10-A	10-G	10-C
11-T	11-T	11-G	11-G	11-C	11-C	11-A	11-A

**Table 2 entropy-25-00476-t002:** Correlation coefficients of adjacent pixel pairs in the original images and their encrypted images.

Image Size	Name		Plain Image			Cipher Image	
		Horizontal	Vertical	Diagonal	Horizontal	Vertical	Diagonal
256×256	Lena	0.9428	0.9143	0.9027	0.0016	−0.0034	−0.0032
	Cameraman	0.9660	0.9357	0.9074	−0.0008	−0.0027	−0.0027
	Peppers	0.9657	0.9410	0.9202	0.0024	0.0019	−0.0016
512×512	4.2.05	0.9689	0.9599	0.9301	0.0027	−0.0011	−0.0011
	4.2.06	0.9724	0.9681	0.9576	−0.0013	−0.0114	−0.0029
	4.2.07	0.9646	0.9615	0.9547	0.0032	0.0018	−0.0011

**Table 3 entropy-25-00476-t003:** *NPCR* and *UACI* values of cipher images.

Image Size	Name	*NPCR*				*UACI*			
		Ref. [5]	Ref. [25]	Ref. [38]	Ours	Ref. [5]	Ref. [25]	Ref. [38]	Ours
256×256		$N_{0.05}^{*}\geq 99.5693$				$U_{0.05}^{*-}=33.2824,U_{0.05}^{*+}=33.6447,{\bar{U}}_{\lambda }^{*}=33.46355$			
	5.1.09	99.603	99.6093	99.5124	99.5712	33.552	33.4723	33.5214	33.4249
	5.1.10	99.636	99.6095	99.6121	99.6094	33.453	33.4663	33.4215	33.5303
	5.1.11	99.942	99.6133	99.5943	99.6262	33.586	33.4554	33.4014	33.4093
	5.1.12	99.792	99.6123	99.5811	99.6109	33.453	33.4604	33.4158	33.4529
	5.1.13	99.792	99.6050	99.5963	99.6292	33.520	33.4601	33.4236	33.5056
	5.1.14	99.6221	99.6110	99.5945	99.6032	33.440	33.4604	33.3951	33.4642
	Mean value	99.731	99.6102	**99.5818**	**99.6084**	33.501	33.4625	33.4298	**33.4645**
	Pass/All	**6/6**	**6/6**	5/6	**6/6**	**6/6**	**6/6**	**6/6**	**6/6**
512×512		$N_{0.05}^{*}\geq 99.5893$				$U_{0.05}^{*-}=33.3730,U_{0.05}^{*+}=33.5541,{\bar{U}}_{\lambda }^{*}=33.46355$			
	5.2.08	99.960	99.6070	99.5858	99.6014	33.692	33.4734	33.3978	33.3901
	5.2.09	99.876	99.6106	99.5812	99.6307	33.548	33.4572	33.4182	33.5037
	5.2.10	99.654	99.6096	99.6100	99.6067	33.454	33.4574	33.4263	33.4822
	7.1.01	99.957	99.6095	99.6028	99.5991	33.648	33.4726	33.4474	33.4482
	7.1.02	99.918	99.6117	99.6078	99.6197	33.465	33.4563	33.4326	33.5738
	7.1.03	99.849	99.6123	99.5811	99.6109	33.273	33.4535	33.4836	33.4847
	7.1.04	99.991	99.6114	99.5946	99.6037	33.202	33.4475	33.4782	33.5274
	7.1.05	99.942	99.6099	99.5937	99.6048	33.830	33.4559	33.4716	33.4679
	7.1.06	99.670	99.6064	99.5912	99.6193	33.627	33.4515	33.4365	33.4049
	7.1.07	99.983	99.6068	99.6014	99.6263	33.609	33.4638	33.4313	33.4707
	7.1.08	99.818	99.6097	99.6013	99.6025	33.375	33.4536	33.4460	33.4628
	7.1.09	99.874	99.6112	99.6148	99.5979	33.530	33.4729	33.3856	33.4370
	7.1.10	99.697	99.6096	99.6097	99.6037	33.438	33.4605	33.3941	33.5011
	boat.512	99.715	99.6084	99.6101	99.5972	33.374	33.4434	33.3973	33.4173
	elaubine.512	99.746	99.6095	99.6185	99.6223	33.379	33.4746	33.4104	33.4945
	gray21.512	99.643	99.6074	99.6034	99.6021	33.507	33.4588	33.4089	33.4351
	numbers.512	99.653	99.6102	99.5941	99.6028	33.388	33.4477	33.4561	33.4904
	ruler.512	99.637	99.6092	99.5945	99.59991	33.415	33.4637	33.4635	33.3932
	Mean value	99.91	99.6095	**99.5998**	**99.6083**	33.486	33.4691	33.4325	**33.4653**
	Pass/All	**18/18**	**18/18**	16/18	**18/18**	12/18	**18/18**	**18/18**	**18/18**
1024×1024		$N_{0.05}^{*}\geq 99.5994$				$U_{0.05}^{*-}=33.4183,U_{0.05}^{*+}=33.5088,{\bar{U}}_{\lambda }^{*}=33.46355$			
	5.3.01	99.950	99.6095	99.6032	99.6024	33.508	33.4511	33.4392	33.4401
	5.3.02	99.982	99.6095	99.6108	99.6057	33.514	33.4536	33.4547	33.4601
	7.2.01	99.980	99.6092	99.6036	99.6109	33.487	33.4606	33.4301	33.4766
	Testpat.1k	99.887	99.6098	99.5971	99.6060	33.453	33.4632	33.4146	33.4638
	Mean value	99.95	99.6095	**99.6037**	**99.6063**	33.491	33.4571	33.4347	**33.4602**
	Pass/All	**4/4**	**4/4**	3/4	**4/4**	**4/4**	**4/4**	**3/4**	**4/4**

**Table 4 entropy-25-00476-t004:** Global Shannon entropy of plain images and cipher images.

Image Size	Name	Plain Images	Cipher Images		
			Ref. [14]	Ref. [25]	Ours
256×256	5.1.09	6.7093	7.9966	7.9971	**7.9973**
	5.1.10	7.3118	7.9971	**7.9974**	7.9973
	5.1.11	6.4523	**7.9975**	7.9969	7.9973
	5.1.12	6.6057	7.9972	7.9972	**7.9974**
	5.1.13	1.5483	7.9965	7.9969	**7.9970**
	5.1.14	7.3424	**7.9977**	7.9974	7.9969
	Best/All		2/6	1/6	**3/6**
512×512	5.2.08	7.5237	7.9991	**7.9993**	**7.9993**
	5.2.09	6.9940	7.9992	**7.9993**	**7.9993**
	5.2.10	5.7056	7.9991	**7.9993**	**7.9993**
	7.1.01	6.0274	7.9990	7.9991	**7.9993**
	7.1.02	4.0045	7.9991	7.9992	**7.9993**
	7.1.03	5.4957	7.9990	**7.9993**	**7.9993**
	7.1.04	6.1074	7.9992	**7.9993**	7.9992
	7.1.05	6.5632	7.9992	7.9992	**7.9993**
	7.1.06	6.6953	7.9992	**7.9993**	7.9992
	7.1.07	5.9916	7.9991	**7.9993**	**7.9993**
	7.1.08	5.0534	7.9990	**7.9973**	**7.9993**
	7.1.09	6.1898	7.9991	7.9992	**7.9994**
	7.1.10	5.9088	7.9990	7.9973	**7.9994**
	boat.512	7.1914	7.9992	**7.9994**	7.9993
	elaubine.512	7.5060	7.9992	**7.9974**	7.9993
	gray21.512	4.3923	7.9993	**7.9994**	**7.9994**
	numbers.512	7.7292	**7.9994**	7.9991	7.9993
	ruler.512	0.5000	7.9987	7.9992	**7.9993**
	Best/All		1/18	11/18	**13/18**
1024×1024	5.3.01	7.5237	**7.9998**	**7.9998**	**7.9998**
	5.3.02	6.8303	7.9996	**7.9998**	**7.9998**
	7.2.01	5.6412	7.9996	**7.9998**	**7.9998**
	Testpat.1k	4.4077	**7.9998**	**7.9998**	**7.9998**
	Best/All		2/4	**4/4**	**4/4**
Total	Best/All		5/28	16/28	**20/28**

**Table 5 entropy-25-00476-t005:** Comparison of local Shannon entropy.

Image Size	Name	Cipher Images		
		Ref. [5]	Ref. [25]	Ours
256×256	5.1.09	7.903369	7.903154	7.902536
	5.1.10	7.903520	7.901680	7.901376
	5.1.11	7.902291	7.902725	7.902147
	5.1.12	7.902721	7.901605	7.902854
	5.1.13	7.902620	7.901269	7.902928
	5.1.14	7.902837	7.902341	7.902519
	Pass/All	4/6	2/6	**5/6**
512×512	5.2.08	7.902793	7.902012	7.902181
	5.2.09	7.902972	7.902484	7.902475
	5.2.10	7.902464	7.902833	7.902317
	7.1.01	7.903339	7.902047	7.902209
	7.1.02	7.902649	7.902568	7.902591
	7.1.03	7.902493	7.902022	7.902006
	7.1.04	7.903261	7.902398	7.902412
	7.1.05	7.902714	7.902568	7.902623
	7.1.06	7.902563	7.902022	7.902171
	7.1.07	7.903185	7.902398	7.902364
	7.1.08	7.902805	7.902137	7.901936
	7.1.09	7.903070	7.902142	7.902964
	7.1.10	7.902929	7.902171	7.902373
	boat.512	7.902697	7.902046	7.902267
	elaubine.512	7.902755	7.902632	7.903213
	gray21.512	7.903661	7.902718	7.901961
	numbers.512	7.902545	7.902067	7.901972
	ruler.512	7.902896	7.902004	7.902361
	Past/All	13/18	**18/18**	17/18
1024×1024	5.3.01	7.902934	7.902057	7.902480
	5.3.02	7.902843	7.902396	7.902249
	7.2.01	7.903238	7.902330	7.902438
	Testpat.1k	7.902715	7.9998	7.9998
	Past/All	3/4	**4/4**	**4/4**
Total	Past/All	20/28	24/28	**26/28**

## Data Availability

Not applicable.

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
