# Peer review of "A Novel Chaotic Image Encryption Scheme Armed with Global Dynamic Selection"

_entropy, 2023, doi:10.3390/e25030476_

Round 1
Reviewer 1 Report
- Please see attachment

Reviewer 2 Report
Authors present an image encryption scheme based on global dynamic selection for DNA encoding rules, using chaos and the well-known permutation and diffusion process. The manuscript is well presented, very complete and detailed. I have only a few comments:
1.- Could you comment the differences in the proposed methodology with respect to those presented in Refs. 3, 18, 24 and 27? Referenced works too uses DNA and Chaos.
2.- In Fig. 1: The output of the second block 2D-LSM is indicate as X, Y. Nevertheless, in the manuscript authors uses the Greek symbol upsilon. Could you be coherent, for a better understanding?
3.- In line 168: The word “and” is repeated (and And).
4.- In line 333: A space before a dot is missing.
